# Forget-me-not! Contrastive Critics for Mitigating Posterior Collapse

**Sachit Menon**[1]        **David Blei**[1]        **Carl Vondrick**[1]

[1]Computer Science Dept., Columbia University, New York, New York, USA

## Abstract

Variational autoencoders (VAEs) suffer from posterior collapse, where the powerful neural networks used for modeling and inference optimize the objective without meaningfully using the latent representation. We introduce *inference critics* that detect and incentivize against posterior collapse by requiring correspondence between latent variables and the observations. By connecting the critic's objective to the literature in self-supervised contrastive representation learning, we show both theoretically and empirically that optimizing inference critics increases the mutual information between observations and latents, mitigating posterior collapse. This approach is straightforward to implement and requires significantly less training time than prior methods, yet obtains competitive results on three established datasets. Overall, the approach lays the foundation to bridge the previously disconnected frameworks of contrastive learning and probabilistic modeling with variational autoencoders, underscoring the benefits both communities may find at their intersection.

## 1 INTRODUCTION

Variational autoencoders (VAEs) provide an integrated approach for simultaneously performing representation learning and generative modeling. Unlike other approaches, such as generative adversarial networks (GANs), VAEs marry the two steps of probabilistic machine learning – inference and modeling – into one framework. They have seen wide success in a number of applications, such as in vision, language, and drug discovery [Kingma and Welling, 2014, 2019].

VAEs posit a very general model, where latent variables $z$ give rise to the data $\mathbf{x}$. The model thus defines the joint distribution $p_\theta(\mathbf{x}, \mathbf{z})$, which factorizes as $p(\mathbf{z})p_\theta(\mathbf{x}|\mathbf{z})$. In this factorization, $p(\mathbf{z})$ corresponds to a prior (for example, a spherical Gaussian), while $p_\theta(\mathbf{x}|\mathbf{z})$ defines an exponential family likelihood (usually a Gaussian) with natural parameter dependent on $\mathbf{z}$. Much of the power of VAEs as generative models comes from how we define this dependence. Typically, we use the powerful function approximation afforded by neural networks to parametrize this relationship.

But in a VAE, the power of neural networks can also be its downfall. With a flexible likelihood, the model can learn to abandon the latents entirely. This allows the approximate posterior, which is also powered by a neural net, to exactly match the prior. This conspiracy of the inference network and the model network allows the VAE to achieve high values for its objective despite both networks forgetting their respective inputs. While we may achieve some generative modeling goals, this *posterior collapse* phenomenon fails at the goal of representation learning [Bowman et al., 2016].

This paper proposes a new approach to mitigate posterior collapse. The key idea is that we can use a *critic* to detect posterior collapse and directly incentivize against it. Consider a set of samples of latent variables and the corresponding observations. If posterior collapse has occurred, corresponding latent/observation are independent. The model is not using the latents, and the approximate posterior just produces independent samples from the prior. On the other hand, if we *are* able to pair up corresponding pairs, they must share some information to allow us to do this, and there is no collapse. With this intuition, we create a critic to accomplish precisely this pairing and integrate it into the VAE objective. The critic constrains the neural network to preserve the mutual information between the latent variables and the observations. The resulting generative model must use the information in the data in its posterior of the latent variables. We call this *forget-me-not regularization*.

Inference critics introduce minimal computational overhead and are easy to train. Unlike other posterior collapse strategies (c.f. [Zhao et al., 2018]), this critic does not require adversarial training. We are not trying to fool the critic

*Accepted for the 38th Conference on Uncertainty in Artificial Intelligence* (UAI 2022).

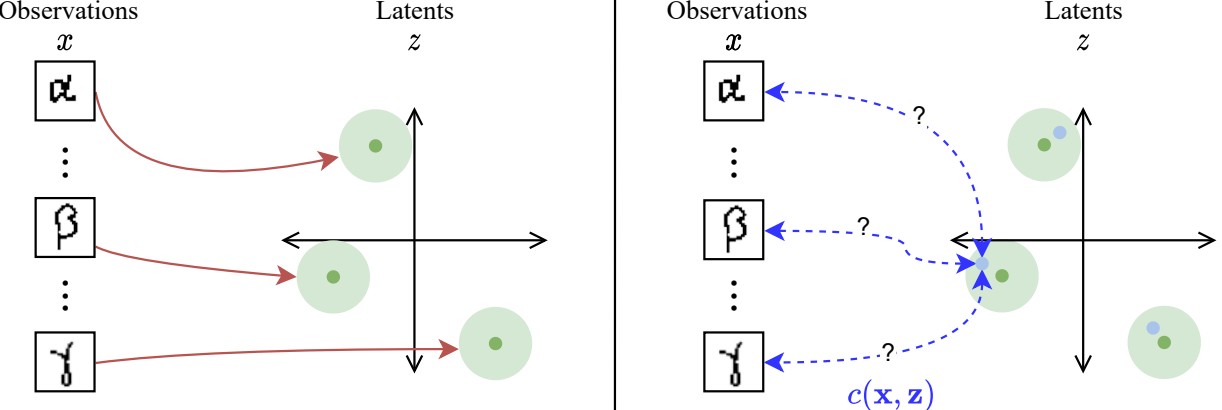

Figure 1: An illustration of the critic. On the left, we have the normal variational network mapping observations to variational parameters (distributions in green). On the right, we show a critic's task for a particular latent sample - it must determine which of the blue arrows marks a true pair.

and have it fail its task of distinguishing corresponding pairs (which would actually encourage posterior collapse). Rather, its loss serves as a regularization, biasing the VAE towards solutions where the latent meaningfully relates to its counterpart in observation space. Moreover, this approach avoids the practical difficulties posed by the 'KL annealing' trick [Bowman et al., 2016], and it does not require multiple experiments to determine a hyperparameter schedule. By connecting the critic to the recent advances in self-supervised contrastive learning [Oord et al., 2016], we show both theoretically and empirically that the inference critic corresponds to increasing the mutual information between the samples and the latents.

Experimental results on three standard datasets (across text and image modalities) show that the inference critic provides a robust strategy for mitigating posterior collapse. The approach is also practical, requiring only minimal computational overhead to the standard VAE. It provides significant efficiency gains over established collapse mitigation strategies while achieving competitive performance.

Our contributions are summarized as: (i) We introduce forget-me-not regularization with inference critics, a self-supervised modification to standard VAEs that substantially reduces effects of posterior collapse. (ii) We show that this modified ELBO formulation directly incentivizes higher mutual information between observations and latents. (iii) We introduce three types of critic: a *neural network critic*, which adds a third neural network to the VAE to act as the critic; a *self critic*, which uses the existing networks to obtain a closed-form optimal solution to the auxiliary task; and a *hybrid critic*, which shares some parameters with the variational network but not all. We contrast these and the effect they have on the final results. (iv) We demonstrate that the method adds less overhead computation time to the standard VAE than other methods for combating posterior collapse.

## 2 POSTERIOR COLLAPSE IN VAES

### 2.1 VAE FUNDAMENTALS

To fit the parameters of a deep generative model, we would ideally maximize the marginal likelihood (the evidence) of the data. However, this is generally an intractable quantity as it involves integrating out the hidden variables. Instead, the most common approach is to use variational inference, which allows us to posit a variational family and maximize a tractable lower bound, the ELBO, over its parameters.

Specifically, the VAE makes use of *amortized* variational inference, which learns a function mapping observations to variational parameters, providing us an approximate posterior over latent variables given observations $q_\phi(\mathbf{x}, \mathbf{z})$. This function, usually parametrized by a neural network with parameters $\phi$, is shared across data points, hence the amortization. This mechanism for amortized inference is also called the 'encoder' in analogy to deterministic autoencoders, with the model referred to as the 'decoder'.

There are many equivalent ways to write the ELBO [Hoffman and Johnson, 2016]. Here, we will focus on a couple that illustrate the problem we are addressing and motivate the approach we propose. First consider:

$$\text{ELBO} = \mathbb{E}_{p_\mathcal{D}(\boldsymbol{x})} \mathbb{E}_{q_\phi(\boldsymbol{z}|\boldsymbol{x})} [\log p_\theta(\boldsymbol{x} \mid \boldsymbol{z})]]] \\ - \text{KL}(q_\phi(\boldsymbol{z} \mid \boldsymbol{x}) \| p(\boldsymbol{z})) \tag{1}$$

where $p_\mathcal{D}(\boldsymbol{x})$ is the empirical distribution of observations from the dataset $\mathcal{D}$. The first term can be thought of as the model conditional likelihood (reconstruction), while the second is the KL divergence between the approximate posterior and the prior.

## 2.2 PITFALLS IN VAE TRAINING

The form of the ELBO in Equation 1 illustrates one reason behind the phenomenon of posterior collapse. If the chosen parametrization of the likelihood is flexible enough to learn to always output (a good approximation of) the data distribution, there is no incentive to take a penalty for the second term: the model can keep the first term high even without letting the approximate posterior deviate from the prior. This is one reason behind the phenomenon of posterior collapse: the model does not need the latent code to maximize the likelihood and thus ignores it.

We can provide another expression for the ELBO that provides insight for this case. Consider the variational joint distribution

$$q_\phi(\mathbf{x}, \boldsymbol{z}) = p_\mathcal{D}(\boldsymbol{x}) q_\phi(\boldsymbol{z} \mid \boldsymbol{x}) \qquad (2)$$

and aggregate posterior

$$q_\phi(\boldsymbol{z}) = \mathbb{E}_{p_\mathcal{D}(\boldsymbol{x})} q_\phi(\boldsymbol{z} \mid \boldsymbol{x}) \qquad (3)$$

where $p_\mathcal{D}(\boldsymbol{x})$ is the data distribution. [Zhao et al., 2018, Hoffman and Johnson, 2016, Dieng et al., 2019, Tomczak and Welling, 2017]. Hoffman and Johnson perform 'ELBO surgery' [Hoffman and Johnson, 2016] to rearrange the ELBO from Equation 1 into the following:

$$\begin{aligned} \text{ELBO} = & \mathbb{E}_{p_\mathcal{D}(\boldsymbol{x})} \mathbb{E}_{q_\phi(\boldsymbol{z}|\boldsymbol{x})} \left[ \log p_\theta(\boldsymbol{x} \mid \boldsymbol{z}) \right] \\ & - \mathcal{I}_q(\boldsymbol{x}, \boldsymbol{z}) - \text{KL}\left( q_\phi(\boldsymbol{z}) \| p(\boldsymbol{z}) \right) \end{aligned} \qquad (4)$$

where $\mathcal{I}_q$ is the mutual information over the variational family. (For a discussion of the variational joint MI $I_q(x; z)$ and the model joint MI $I_p(x; z)$, see Appendix A.)

In other words, the KL divergence between the posterior and the prior decomposes into a mutual information (MI) penalty and a KL term that encourages matching the aggregate posterior and the prior. Maximizing the ELBO thus explicitly discourages high mutual information between observations and latents, pushing them towards independence. As [Hoffman and Johnson, 2016] show, decreasing the value of the MI does not impact the likelihood term, which tends to dominate. As such, the objective enables the flexible neural nets to achieve solutions exhibiting posterior collapse.

This loss of information means that the latents will not be informative about the observations (and thus cannot be useful representations). In the next section, we introduce a method to explicitly prevent this information loss or 'forgetting'. By incorporating a regularization term that incentivizes higher MI into the ELBO, we will counteract the effect of the MI penalty from Equation 4.

## 3 FORGET-ME-NOT REGULARIZATION

We will employ a critic that imposes a penalty on the objective if observations and their corresponding latents cannot be distinguished from non-corresponding pairs. The intuition is that this matching is only possible if there is information shared between observations and latents.

## 3.1 THE INFERENCE CRITIC

Consider a batch of samples from the empirical data distribution $x_0, \ldots, x_k$, and a corresponding batch of latent samples $z_0, \ldots, z_k$ (by encoding the $x_i$). Every pair with the correct correspondence comes from the variational joint distribution $q(\mathbf{x}, \mathbf{z})$, while non-corresponding pairs are independent and come from the product of marginals ($\mathbf{z}$ via ancestral sampling) [Alemi et al., 2018]. Formally:

$$(z_i, x_j) \sim \begin{cases} q_\phi(\mathbf{z}, \mathbf{x}) & i = j \\ q_\phi(\mathbf{z}) p_\mathcal{D}(\mathbf{x}) & i \neq j \end{cases} \qquad (5)$$

If we can distinguish the joint distribution from the product of marginals, there must be some dependence between $\mathbf{x}$ and $\mathbf{z}$. Given samples from both distributions, the critic will try to pick out which pairs belong to which distribution. The more successful it is, the more different the distributions must be, and therefore the more $\mathbf{x}$ and $\mathbf{z}$ must be related.

The classifier, which we will denote $f$, needs to distinguish between the joint and the product of marginals by directly *contrasting* the correct pairings from the incorrect ones. We know that for every observation $x_i$, there is one latent in the batch $z_i$ that corresponds to it (and vice versa). We also know that the others *should not*. This corresponds to softmax classification [Bishop, 2006], or a probabilistic classifier with a categorical likelihood. $f$ maximizes the objective:

$$c(\mathbf{x}, \mathbf{z}) = \mathbb{E}\left[ \log \frac{f(x^+, z^+)}{\sum_{x \in X} f(x, z^+)} \right] \qquad (6)$$

which is the critic's expected value for the true corresponding pairs (denoted by the $+$) relative to all the other (non-corresponding) pairs, across pairs.[1] (The notation within the sum is in reference to a particular positive $z^+$; we can think of it as considering a particular $z^+$ and trying to find the associated $x^+$ among the options for $x$, then taking the expectation over $z$. This is symmetric with respect to choosing an $x^+$ and finding the associated $z^+$.)

Crucially, this objective constitutes a lower bound of the mutual information, shown by Oord et al. [2019] in the context of self-supervised representation learning. By maximizing Equation 6, the classifier approximates the density ratio between the joint distribution and product of marginals [Oord et al., 2019, Song and Ermon, 2020], which is precisely the ratio appearing in the mutual information. Therefore, by optimizing the parameters of the VAE with this objective added to the ELBO as a regularizer, we push up on this

---

[1]Note writing $f(x, z)$ as $\exp(f_0(x, z))$ recovers the softmax classifier exactly.

**Algorithm 1:** Forget-me-not regularization with neural network inference critic.

**Input:** Dataset $\mathcal{D}$, batch size $K$, initial VAE parameters $\theta$, initial critic $f$ parametrized by $\psi$, regularization weight $\lambda$

1 **while** *not converged* **do**
2   Sample from $p_{\mathcal{D}}(\mathbf{x})$ $K$ times to obtain $(x^{(i)})_{i=1}^{K}$
3   Sample $z^{(i)} \sim q_{\phi}(\mathbf{z}|x^{(i)}) \forall i \in \{1, \ldots, K\}$
4   Compute $\mathcal{L}_0 = \sum_{i=1}^{K} \text{ELBO}(\theta, \phi, x^{(i)}, z^{(i)})$ per Eqn 1 ;   // standard minibatch ELBO
5   Compute $f_{\psi}(x_i, z_j) \forall i, j$ ;    // inference critic values
6   $\mathcal{L}_1 \leftarrow c(\mathbf{x}, \mathbf{z})$ per Eqn 6 ;    // inference critic objective
7   $\mathcal{L} \leftarrow \mathcal{L}_0 + \lambda * \mathcal{L}_1$
8   Perform gradient update for VAE parameters
9   Perform gradient update for critic parameters

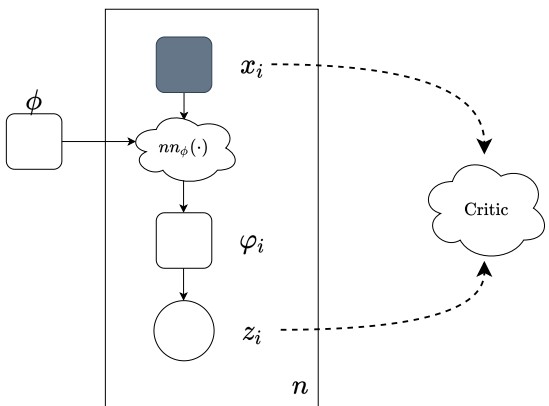

Figure 2: Forget-me-not regularization in reference to the graphical models acted on. Dashed lines show computation flow. The inference critic works against information loss in the variational family, trying to ensure it can tell $z_i$ and $x_i$ correspond. It does this by solving a classification task: among a batch of different repetitions of these variables, classify which belong together. This task can only be solved if there is shared information between variables belonging to the same subgraph (repeated by the plate) to distinguish them from those belonging to others.

lower bound. This push increases the MI between the latents and the observations, effectively mitigating posterior collapse as desired.

This critic establishes a tight connection to the contrastive learning literature, in particular the InfoNCE loss from Contrastive Predictive Coding (CPC) [Oord et al., 2019], enabling the many advances in self-supervised learning to be applied to VAEs. See Appendix D for details of how this connection and bound apply. Note that, in the MI estimation literature, classifiers that estimate the mutual information like CPC are sometimes also referred to as critics [Poole et al., 2019]. We intentionally overload this word here: the inference critics of this paper critique the variational inference optimization.

### 3.2 REGULARIZATION

In order to prevent posterior collapse, we want to maximize the mutual information between the latents and the samples. We integrate the penalty of the critic, which aims to maximize $c(\mathbf{x}, \mathbf{z})$, to the ELBO:

$$\begin{aligned} \text{ELBO}_{\text{CRITIC}} = \; & \mathbb{E}_{p_{\mathcal{D}}(\boldsymbol{x})} \mathbb{E}_{q_{\phi}(\boldsymbol{z}|\boldsymbol{x})} \left[ \log p_{\theta}(\boldsymbol{x} \mid \boldsymbol{z}) \right] \\ & - \text{KL} \left( q_{\phi}(\boldsymbol{z} \mid \boldsymbol{x}) \| p(\boldsymbol{z}) \right) + c(\mathbf{x}, \mathbf{z}) \end{aligned} \quad (7)$$

We optimize the parameters for the variational family, model, and critic jointly; the critic prevents the usual conspiracy between the model and variational family over the course of training, avoiding collapse. Algorithm 1 illustrates the training procedure.

Notice that the mutual information appears in both the KL-divergence term and the critic term. When the true mutual information is equal to our estimate, this corresponds to matching the marginals for $z$ and the mutual information

term from Equation 4 is cancelled out by the regularization. This approach does not need to resort to adversarial training techniques that are difficult to use in practice. Instead, the critic provides a straightforward mechanism to mitigate posterior collapse by causing an increase in the mutual information term.

We apply this regularization across the variational family - an *inference* critic, illustrated in Figure 2. See Appendix E for discussion of its potential application across the model. Furthermore, while here we consider the problem of posterior collapse by examining the ELBO for the VAE, posterior collapse is a more general phenomenon, observed in many different models when using amortized variational inference with deep neural networks. In theory, this approach would work on any such problem with amortized inference.

### 3.3 TYPES OF INFERENCE CRITICS

This framework affords multiple types of critics (as forms for $f$ in Equation 6) that correspond to the mutual information between the latents and the samples. We propose three types of inference critics.

The most straightforward critic is a **neural network critic**. This critic uses a third network, entirely separate from those used in the VAE, to implement the critic. The choice of the network design depends on the structure of the modality. For example, we could use an embedding layer followed by an LSTM for text data.

We also propose a **self-critic** that uses the variational family as its own critic, providing the tightest estimate of the mutual information. This originates from an idea in noise contrastive estimation: if we have a tractable conditional (as is the case with the variational family in the VAE), we can directly use it to estimate the mutual information, and in particular the tightest estimate of the MI will be found by using $f(x, z) = \log q(z|x)$ [Poole et al., 2019]. This formulation incentivizes the log-likelihood under the conditional to be highest for samples that actually belong together, as we would desire. This critic also has the advantage of not requiring additional parameters.

The **hybrid critic**, as the name suggests, is in-between the self-critic and the neural network critic. Rather than use an entirely separate neural network, this critic shares early layers with the variational network, after which point it has its own parameters. This approach can be well-suited to text data, where we may wish to share the embedding weights between the variational network and critic but have them separate past that. This presents a way to compromise between using no additional parameters (as in the self-critic) and using a whole additional network's worth of parameters (as in the neural network critic).

# 4 EXPERIMENTS

The basic objective of our experiments is to analyze inference critics under posterior collapse. We report results across three established image and text datasets.

## 4.1 COMMON EXPERIMENTAL SETUP

We follow a common setup throughout our experiments. Since the method is compatible with the VAE family, it can also be added on top of existing methods to mitigate posterior collapse. We conducted experiments adding forget-me-not regularization to a standard VAE to assess whether the theoretical improvements yielded empirical benefits, following [He et al., 2019] and collapse metrics from [Dieng et al., 2019]. We measure measuring the approximate negative log-likelihood (NLL, via 500 importance samples - this gives a tighter bound for the evaluation than the ELBO), the ELBO's KL term, a Monte Carlo estimate of the MI under the variational joint $I_q(x; z)$, and the number of active units (AU) [Burda et al., 2016] on held-out data.

We report results for the Yahoo, Yelp, and Omniglot datasets, allowing systematic comparison to prior work [He et al., 2019, Kim et al., 2018, Dieng et al., 2019]. As there are both visual and text datasets, we use existing, appropriate neural network architectures for each modality, which we describe next to each experiment. For all datasets, we follow the standard train/validation splits provided by the original dataset authors. We evaluated all results on a single NVIDIA

GeForce RTX 2048 GPU.

**Baselines:** For all experiments, we compare against multiple established baselines. We use the standard **VAE [Kingma and Welling, 2014]** without any additional strategy for handling posterior collapse. We also compare to **SA-VAE [Kim et al., 2018]**. Rather than solely relying on amortized inference to obtain variational parameters, the semi-amortized VAE (SA-VAE) uses amortized inference to obtain an initialization and updates the parameters directly from that point. Finally, we compare to the **Lagging-VAE [He et al., 2019]**, which aggressively updates the variational family many more times (e.g. 50x) as frequently as the model. We choose this as representative as Lagging-VAE holds the previous state-of-the-art on both modeling and posterior collapse metrics without requiring use of the KL annealing trick, making it a competitive baseline.

## 4.2 EVALUATION METRICS

We evaluate all models and baselines using the standard metrics for evaluating the posterior collapse of VAEs. We report the following:

**Negative Log Likelihood (NLL):** The negative log-likelihood indicates the modeling performance on held-out data. A smaller NLL indicates that the model generalizes to data samples well.

**KL-Divergence (KL):** The KL-divergence term of the ELBO in Equation 1 is a commmon indicator of collapse. If we obtain a good ELBO but the KL term is low, the optimization progress comes from the model likelihood (the first term of Equation 1). In this case, especially if the KL is at or near zero, the posterior matches the prior too well (suggesting collapse has occurred).

**Mutual Information (MI):** The estimate of the mutual information across the variational family, $I_q(x; z)$, aims to estimate if the latents have become independent of the data (over the variational joint). We compute this estimate as the difference of the previously-described KL term and the marginal KL term from equation 4 per Hoffman and Johnson [2016]. Both KL terms are obtained by Monte Carlo. The first is obtained naturally in computation of the ELBO, while the second can be computed via ancestral sampling (sampling from the dataset, then the approximate posterior) again as in [Hoffman and Johnson, 2016]. However, as pointed out by [He et al., 2019], this estimate is biased - specifically, it is an upper bound.

**Active Units (AU):** A standard metric from prior work, the number of active units provides a measure of how many latent dimensions are active, which is specifically how many of the stochastic units show any variation when the input varies. If few are active, we are likely collapsed. Activity is measured by $A_z = \mathrm{Cov}_{\mathbf{x}}\left(\mathbb{E}_{z \sim q(z|\mathbf{x})}[z]\right)$, with a unit

| | Yahoo | | | | Yelp | | | |
|---|---|---|---|---|---|---|---|---|
| Model | NLL | KL | MI ($I_q$) | AU | NLL | KL | MI ($I_q$) | AU |
| VAE | 328.9 (0.1) | 0.0 (0.0) | 0.0 (0.0) | 0.0 (0.0) | 358.3 (0.2) | 0.0 (0.0) | 0.0 (0.0) | 0.0 (0.0) |
| SA-VAE [Kim et al., 2018] | 329.2 (0.2) | 0.1 (0.0) | 0.1 (0.0) | 0.8 (0.4) | 357.8 (0.2) | 0.3 (0.1) | 0.3 (0.0) | 1.0 (0.0) |
| Skip-VAE [Dieng et al., 2019] | 328.7 (0.3) | 0.22 (0.1) | 0.0 (0.0) | 7.0 (0.6) | 358.1 (0.3) | 0.15 (0.0) | 0.0 (0.0) | 4.6 (0.5) |
| Lagging-VAE [He et al., 2019] | **328.2** (0.2) | 5.6 (0.2) | 3.0 (0.0) | 8.0 (0.0) | **356.9** (0.2) | 3.4 (0.3) | 2.4 (0.1) | 7.4 (1.3) |
| VAE + Inference Critic (Self) | 328.7 (0.2) | 3.6 (0.1) | 2.6 (0.0) | 3.0 (0.0) | 358.2 (0.2) | 3.8 (0.1) | 2.7 (0.0) | 3.0 (0.0) |
| VAE + Inference Critic (Hybrid) | **328.2** (0.1) | 4.3 (0.1) | 2.8 (0.0) | **11.0** (0.4) | 357.7 (0.2) | 4.0 (0.1) | 2.8 (0.0) | 7.0 (0.0) |
| VAE + Inference Critic (NN) | 338.9 (0.4) | **17.5** (1.1) | **3.3** (0.0) | 8.0 (1.0) | 370.5 (0.5) | **18.6** (1.8) | **3.2** (0.1) | **12.0** (2.0) |

Table 1: Quantitative results on the Yahoo and Yelp text corpora. Each critic improves on collapse metrics when added to the standard VAE with no other changes. Results for comparison without KL annealing were referenced from He et al. [2019] or re-implemented in the same framework and are averages of 5 runs, with standard deviation given in parentheses. (We follow the training details in the aforementioned methods, running until convergence is achieved on the validation ELBO.)

| | Omniglot | | | |
|---|---|---|---|---|
| Model | NLL | KL | MI ($I_q$) | AU |
| VAE | 89.41 (0.04) | 1.51 (0.05) | 1.43 (0.07) | 3.0 (0.0) |
| SA-VAE [Kim et al., 2018] | 89.29 (0.04) | 2.55 (0.05) | 2.20 (0.03) | 4.0 (0.0) |
| Skip-VAE [Dieng et al., 2019] | 89.41 (0.05) | 1.75 (0.20) | 1.61 (0.10) | 3.0 (0.4) |
| Lagging-VAE [He et al., 2019] | **89.05** (0.05) | 2.51 (0.14) | 2.19 (0.08) | 5.6 (0.5) |
| VAE + Inference Critic (Self) | 89.18 (0.04) | 6.30 (0.12) | 3.75 (0.04) | 11.0 (1.0) |
| VAE + Inference Critic (Hybrid) | 89.16 (0.04) | 6.41 (0.15) | 3.78 (0.03) | 13.0 (0.7) |
| VAE + Inference Critic (NN) | 89.24 (0.05) | **7.66** (0.14) | **3.82** (0.04) | **28.0** (0.0) |

Table 2: Quantitative results on the Omniglot image dataset. We find each critic improves on collapse metrics when added to the standard VAE with no other changes. Results for comparison without KL annealing were referenced from [He et al., 2019] or re-implemented in the same framework and are averages of 5 runs, with standard deviation given in parentheses. (We follow the training details in the aforementioned methods, running until convergence is achieved on the validation ELBO.)

| | Running Time | |
|---|---|---|
| Model | Factor | Absolute (Hrs) |
| VAE | 1.00 | 3.5 |
| Lagging-VAE | 3.6 | 12.7 |
| VAE + Inf. Critic (Hybrid) | 1.06 | 3.7 |

Table 3: Speed comparison results. We report wall clock time ('Absolute') and factor increase over the baseline ('Factor') on the Yahoo benchmark. Adding an inference critic adds minimal overhead. Per the experimental procedure used in [He et al., 2019], we run to convergence on the validation ELBO; the standard VAE converges after 54 epochs, Lagging-VAE converges after 49 epochs, and the VAE with a hybrid critic converges after 54 epochs. For the same number of epochs, the speed difference only increases.

considered active if its activity is above some threshold (we follow [Dieng et al., 2019] and [He et al., 2019] with a threshold of 0.01).

## 4.3 RESULTS ON TEXT

**Experimental Setup:** In this experiment, we evaluate on the Yahoo and Yelp benchmarks [Yang et al., 2017]. All methods use the standard train/val/test splits and follow the experimental protocol in [He et al., 2019], fully described in Appendix F. For the neural network architecture, this is a 1-layer LSTM with learnable embeddings for the variational network and for the model network. Following this protocol, all methods use a 32-dimensional latent space and a batch size of 32.

**Quantitative Results:** The quantitative results in Table 1 show that all critics substantially improve on the collapse metrics compared to the baseline on both datasets, showing forget-me-not regularization is able to significantly mitigate posterior collapse on text data.

Our results show that different critics have different behaviors. As the self-critic optimally solves the auxiliary task at each step, if there is any information shared between observations and latents, it is able to pair them up successfully. On the other hand, the neural network critic is entirely separate from the variational network and is decidedly sub-optimal at the auxiliary task - and thus needs more information to be shared, as that increases its chances of finding a way to tie together corresponding pairs. As the experimental results show, this encourages less collapse. In particular, the neural network critic reaches close to the theoretical maximum $I_q$ increase offered by applying an inference critic with the batch size used ($\log(32) \approx 3.4$, see Appendix D). At the same time, this may lead to solutions that do not perform as well along the NLL, because there is a much stronger pull

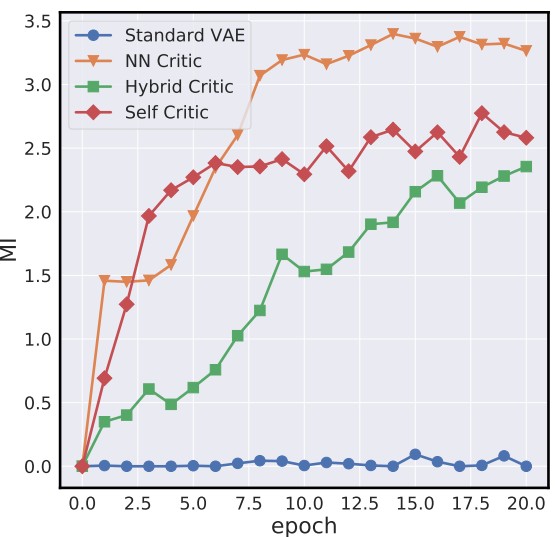

Figure 3: Comparison of mutual information across the variational family ($I_q$) for various critics vs baseline over the first 20 epochs of training on the Yahoo benchmark. This is cropped for clarity; the plot over the entirety of training can be seen in Appendix G.

to not collapse. The hybrid critic reaches a balance between these: by not being optimal at solving the auxiliary task, it pulls more strongly away from collapse, but by sharing some parameters with the variational network, it is able to reach better modeling solutions than the entirely separate neural network critic, actually improving the NLL slightly over the standard VAE. It is interesting that the neural network critic, which uses an entirely separate set of parameters and thus should be strictly more flexible, fails to reach such an optima; we hypothesize this comes from difficulty in optimization, which is made easier when the weights are tied. None of the critics require additional hyperparameter scheduling, such as to control KL annealing.

Figure 3 shows that all critics progressively push the mutual information up over the course of training. In contrast, the standard VAE remains collapsed throughout training.

### 4.4 RESULTS ON IMAGES

**Experimental Setup:** We next evaluate on the Omniglot benchmark [Lake et al., 2015] with the provided train/val/test split. All methods follow the experimental protocol in He et al. [2019], fully described in the Appendix. For the architecture, all methods use a ResNet [He et al., 2015] for the variational network and a 13-layer Gated PixelCNN [Oord et al., 2016] for the model. All methods use a 32-dimensional latent space with a batch size of 50.

**Quantitative Results:** The results on images in Table 2 show a similar trend to the results on text, suggesting that the approach is robust across modalities. We find inference critic is able to mitigate collapse for images as well, improving

substantially on collapse metrics compared to the baseline.

Furthermore, the relative quantitative behavior of different critics remains the same on the images as well as text. In particular, the neural network critic reaches close to the theoretical maximum $I_q$ increase offered by applying an inference critic with the batch size used ($\log(50) \approx 3.9$. See Appendix D) for more details. Finally, unlike for text, all critics are able to improve NLL over the baseline. It is interesting that the regularization actually seems to help the original objective; this behavior is unlike that of approaches like Beta-VAE [Higgins et al., 2016]. We hypothesize that while the collapsed solution may be 'easy' to arrive to with SGD, there exist better optima that actually use the latents, which inference critics drive the parameters towards.

### 4.5 EFFICIENCY AND RUNNING TIME PERFORMANCE

A key advantage of this approach is that it adds minimal overhead to standard VAE training. As Table 3 shows, the hybrid critic completes training in only 1.06x the time the standard VAE takes - 3.7 hours vs 3.5 hours in wall-clock time on one NVIDIA RTX 2048. This comes at no performance trade-off: the hybrid critic improves along the collapse metrics KL, MI, and AU over the standard baseline with no reduction in NLL. The time performance is particularly important as the overhead of previous approaches to avoiding posterior collapse is typically too high for large-scale problems. While Lagging-VAE does help against posterior collapse, it took 12.7 hours to train (3.4x the time of the hybrid critic). Other approaches are reported in the literature to require further computation - for example, SA-VAE takes between 4 and 8 times as long as Lagging-VAE in [He et al., 2019].

How can our method have such low overhead? Contrastive learning often can be quite expensive. The 6% overhead is made possible with the hybrid critic, as it shares most of its parameters with the inference network. The bulk of the computation is thus only done once, with the overhead introduced when the two diverge. Additionally, with inspiration from current work in self-supervised learning we employ critics that are separable [Poole et al., 2019], meaning they do not need to take every $(x, z)$ pair (which would be $n^2$ forward passes) but only the x. This leads to quadratic big-O speedup compared to other approaches that need to process every potential pairing.

## 5 RELATED WORK

The most relevant line of study related to this work are methods that try to avoid posterior collapse - especially those that do so by increasing the mutual information. [Bowman et al., 2016] identify the problem of posterior collapse for VAEs

endowed with powerful generators. They prescribe 'KL annealing', or slowly increasing the KL-penalty in Equation 1 at the start of training. From Equation 4, we can interpret this as slowly ramping up the MI penalty inherent to the ELBO. Approaches like [Chen et al., 2017], [Gulrajani et al., 2016], [Yang et al., 2017] modify the architecture of the generative model to reduce its flexibility, with the hope that this will prevent it from finding solutions that 'forget'. [Dieng et al., 2019] instead adds skip connections to the VAE model network to increase $I_p(x; z)$.

Other approaches aim to explicitly encourage higher mutual information by modifying the objective, towards the goal of 'fixing a broken ELBO' as identified by Alemi et al. [2018]. Forget-me-not regularization falls into this category. (Note that we do not claim novelty in the general idea of modifying the objective to avoid MI loss; rather, forget-me-not regularization presents a new way of accomplishing this with distinct advantages in optimization ease, simplicity, and speed.) Zhao et al. [2018] introduce an explicit marginal-KL penalty which can be traded off with the usual KL term in Equation 1. They do this either with an adversarial classifier that tries to guess if a sample is from the aggregate posterior or the prior, Stein variational gradient descent [Liu and Wang, 2019], or kernel-based methods (e.g., MMD [Gretton et al., 2012]), each of which is difficult to work with. They show adversarial autoencoders [Makhzani et al., 2016] are a special case of the former. raz [2019] enforce a minimum KL instead of using a penalty.

Wasserstein autoencoders [Tolstikhin et al., 2019] provide another approach to marginal matching, aiming to avoid the traditional KL term (and its associated MI penalty) entirely by matching marginals with the Wasserstein distance (but this also requires adversarial training or MMD - it is also a generalization of adversarial autoencoders). Phuong et al. [2018] adds an explicit penalty for the MI with the Barber and Agakov lower bound [Barber and Agakov, 2004], but this is not fully differentiable and requires use of the high-variance REINFORCE gradient estimator. Closely related to this work is Rezaabad and Vishwanath [2020], which also uses an auxiliary network on batches of samples (from the variational joint) to obtain a penalty term; instead of solving a straightforward classification problem as done here, they use these to try to estimate the dual form of the mutual information. VAE-MINE [Qian and Cheung, 2019] use the MINE bound (a cousin of the CPC bound) to aim to increase MI, but as pointed out by Poole et al. [2019], the way it is computed does not constitute a correct MI bound. (See Appendix H for more detailed discussion of VAE-MINE.) Along different lines, Kim et al. [2018] address posterior collapse by using the inference network's outputs as an initialization for SVI, creating a hybrid procedure between amortized and non-amortized inference. He et al. [2019] modify the optimization procedure to take more steps for the inference network than the model network.

The idea of including an auxiliary classification task to generative models has a long history. The most famous of these are GANs [Goodfellow et al., 2014], which train an auxiliary classifier adversarially on the empirical data distribution and the model data distribution, aiming to make these indistinguishable. Uehara et al. [2016] interprets this in the framework of density ratio estimation. Bayesian GANs Tran et al. [2017] modify these with Bayesian neural networks for Bayesian inference. Hybrids between VAEs and GANs have been introduced in many forms [Larsen et al., 2016, Donahue et al., 2017, Srivastava et al., 2017, Mescheder et al., 2017]. All of these use the auxiliary classifier adversarially, to encourage distribution matching, whereas we are encouraging distributions *not* to match - specifically, we are encouraging dependence between latents and observations by making the joint different from the product of marginals. ane [2021] use an auxiliary classifier for noise-contrastive estimation between the prior and the aggregate posterior to address the 'prior hole' problem. Rather than auxiliary classifiers, sey [2019] add auxiliary decoders aiming to avoid collapsed optima.

This work provides a foundation for connecting advances in contrastive learning with the VAE framework, giving us control over *what* information is preserved (vs what the representations are invariant to). CPC [Oord et al., 2019] introduces the InfoNCE mutual information bound, applying it to representations obtained by applying autoregressive models to sequential data. [Wu et al., 2018] proposes the instance discrimination problem, which suggests matching observations with encodings of perturbed versions of the same observation - for example, two different augmentations of an image, like brightness shifts - learning representations that are invariant to these factors. MoCo [He et al., 2020] proposes a 'momentum queue' to hold encodings from recent batches to increase the number of 'negatives' being compared to substantially; it surpasses the performance of representations learned by supervised neural networks on various computer vision tasks. Zhu et al. [2020] allows for contrastive learning without large numbers of negatives via a small modification to the InfoNCE objective, that provides a more practical bound on the mutual information. Tschannen et al. [2020] provides connections of these techniques and associated mutual information bounds to metric learning, which could also be an interesting perspective to bring to representation learning with VAEs.

# 6 CONCLUSIONS

We present a new method for protecting against posterior collapse in VAEs. In doing so, we establish a connection between VAEs and contrastive representation learning. We show inference critics increase the mutual information between latents and observations by maximizing the CPC lower bound. Experiments on three datasets show the effectiveness of the approach, with significant efficiency gains.

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
