# OpenReview forum: "Forget-me-not! Contrastive Critics for Mitigating Posterior Collapse"
_auai.org/UAI/2022/Conference — UAI 2022 Poster_

### Official Review · Reviewer_NuC8 · 2022-04-09

**Q2(1) Originality/Novelty:** 1
**Q2(2) Significance/Impact:** 2
**Q2(3) Correctness/Technical Quality:** 3
**Q2(6) Clarity Of Writing:** 4
**Q6 Overall Score:** 4
**Q8 Confidence In Your Score:** 4

**Q1 Summary And Contributions:**

This paper proposes to use self-supervised learning to mitigate posterior collapse, where the self-supervised constrastive loss essentially maximize the mutual information between input and latent.

**Q2 Assessment Of The Paper:**

More detailed information regarding each of these aspects is given below:

**Q2(4) Quality Of Experiments (Optional):**

3: Good: The experimental evaluation is adequate, and the results convincingly support the main claims.

**Q2(5) Reproducibility:**

3: Good: Key resources (e.g., proofs, code, data) are available and key details (e.g., proofs, experimental setup) are sufficiently well-described for competent researchers to confidently reproduce the main results.

**Q3 Main Strengths:**

1. The proposed method is straightforward and easy to implement.
2. The authors utilize multiple evaluation metrics to fully demonstrate the effectiveness of the proposed method.
3. Experiments on both text and image demonstrate the superiority of the proposed method.

**Q4 Main Weakness:**

1. limited novelty: using MI to mitigate posterior collapse has been studied before as in VAE-MINE (already cited in related work), so the only different part here is a different MI estimator, which itself is also not a novel contribution.

**Q5 Detailed Comments To The Authors:**

1. missing citation of other works that combine VAE and constractive learning [1].
2. missing citation and comparison with other methods that mitigate posterior collapse [2].

If I understand it correctly, posterior collapse results fromthe use of a powerful decoder so that it ignores latent code, so why not maximizing the MI between latent and decoder output?



[1] Aneja J, Schwing A, Kautz J, et al. A contrastive learning approach for training variational autoencoder priors[J]. Advances in Neural Information Processing Systems, 2021, 34.
[2] Razavi A, Oord A, Poole B, et al. Preventing posterior collapse with delta-vaes[J]. arXiv preprint arXiv:1901.03416, 2019.

**Q7 Justification For Your Score:**

The proposed method does not have enough novel contribution.

**Q9 Complying With Reviewing Instructions:**

1: Yes.

---

### Official Review · Reviewer_3NDi · 2022-04-10

**Q2(1) Originality/Novelty:** 2
**Q2(2) Significance/Impact:** 3
**Q2(3) Correctness/Technical Quality:** 3
**Q2(6) Clarity Of Writing:** 3
**Q6 Overall Score:** 6
**Q8 Confidence In Your Score:** 3

**Q1 Summary And Contributions:**

VAEs suffer from posterior collapse, where the decoder learns to model the log-evidence rather than the conditional likelihood. That is, the model ignores the dependency between the latent variable and the observed variables. The ELBO, used in VAEs, is known to have several optima, one of them being this degenerated case.

This work proposes to use contrastive learning as a penalty term to steer the training towards the desired optima that utilizes information from the latent variables.

**Q2 Assessment Of The Paper:**

More detailed information regarding each of these aspects is given below:

**Q2(4) Quality Of Experiments (Optional):**

3: Good: The experimental evaluation is adequate, and the results convincingly support the main claims.

**Q2(5) Reproducibility:**

2: Fair: Key resources (e.g., proofs, code, data) are unavailable but key details (e.g., proof sketches, experimental setup) are sufficiently well-described for an expert to confidently reproduce the main results.

**Q3 Main Strengths:**

- The motivation is clear and the proposed idea simple to understand.
- The paper leverages ideas from self-supervised learning to improve VAEs, giving room to future improvements.
- Experimentation is good enough to support the claims made, giving a lightweight alternative to Lagging-VAE.



**Q4 Main Weakness:**

- 4.1 I find section 3.1 a bit forced and unnatural, where the connection with self-supervised learning is really clear already at the end of section 2.
- 4.2 Presentation could be improved. There are several typos (e.g. Eq 5), and figures need quite an improvement (e.g., Figure 2 is cutted and not referenced).
- 4.3 I do not fully understand why the sum on Eq. 6 is over X, rather than all pairs (X, z), as in contrastive learning. This is not discussed.

Experiments:
- 4.4 I do not undestand how the overhead of the method is only of 6%. Contrastive learning is known for being quite expensive. Moreover, it is also known for heavily depend on the batch size and the size of the denominator's set, but there is no ablation study.
- 4.5 While the metrics of Dieng et. al. are used, their method is not compared.
- 4.6 Results do not show standard deviation and no statistical test is performed.
- 4.7 Figure 6 does not show the same as Table 1 for the Hybrid approach (why only 20 epochs?)



**Q5 Detailed Comments To The Authors:**

- Typo in Eq. 1.
- Page 2: Our conclusions -> Our contributions.
- The way equations are cited is inconsistent.
- Algorithm's line numbers are in the margin.
- Figure 2 is not referenced, and the space between caption and the main text is minuscule.
- It would be nice to describe the limitations of your work and future work.

**Q7 Justification For Your Score:**

The idea is simple and good, and connections with SSL are interesting. However, the paper would benefit a lot of more work on the presentation and the experiments (which are not described either, but deferred to other works).

However, I think the pros outweigh the cons.

====

After the rebuttal, I trust the authors in implementing the promised changes, and I have therefore updated my score.

**Q9 Complying With Reviewing Instructions:**

1: Yes.

---

### Official Review · Reviewer_ixUE · 2022-04-13

**Q2(1) Originality/Novelty:** 2
**Q2(2) Significance/Impact:** 2
**Q2(3) Correctness/Technical Quality:** 3
**Q2(6) Clarity Of Writing:** 4
**Q6 Overall Score:** 5
**Q8 Confidence In Your Score:** 3

**Q1 Summary And Contributions:**

The authors propose a simple method for preventing posterior collapse in VAEs. In particular, they augment the VAE objective by a contrastive loss term that ensures that latent variables and obervations share some information, thereby avoiding the collapse. They demonstrate in experiments that their approach compares favorable to severable baselines with respect to several measures.

**Q2 Assessment Of The Paper:**

More detailed information regarding each of these aspects is given below:

**Q2(4) Quality Of Experiments (Optional):**

3: Good: The experimental evaluation is adequate, and the results convincingly support the main claims.

**Q2(5) Reproducibility:**

3: Good: Key resources (e.g., proofs, code, data) are available and key details (e.g., proofs, experimental setup) are sufficiently well-described for competent researchers to confidently reproduce the main results.

**Q3 Main Strengths:**

Considers an important problem and proposes an easy solution which shows some favorable results in experiments.

Mainly well written and easy to follow.

The proposed method is novel to the best of my knowledge. However, the problem considered is not new and there are already some existing "solutions".



**Q4 Main Weakness:**

As far as I saw no source code is part of submission and the authors do not say whether they will release their code. This can limit reproducibility. On the other hand the method appears easy enough to be reliably implemented by oneself.

The problem is relevant - thus I expect some impact but the empirical evaluation might not be sufficient to convince a broader audience.

**Q5 Detailed Comments To The Authors:**

As said, I like the simple but effective proposal made in the paper. The paper could be improved by extending the discussion of related work (see also below) and extending the empirical comparison as well. The discussion of the different types of critics and their analysis should be extended (e.g., how do parameter choices of the NN critic affect performance; why is the difference in performance between the critics sometimes large, sometimes negligible). Furtermore, I would be curious to understand better why the proposed critics in most cases do not achieve the best generative performance (lowest NLL) - it seems that we are trading-off generative performance with mutual information but to me it is not intuitively clear why this is the case (related to that: I don't agree with the statement "This comes at no perfor- mance trade-off: the hybrid critic improves along NLL, KL, MI, and AU over the standard baseline."; this should be toned down and probably be rephrased as "no reduction in NLL").

Some relevant citations and the discussion of the respective methods are missing, e.g.
* Razavi, Ali, et al. "Preventing Posterior Collapse with delta-VAEs." International Conference on Learning Representations. 2018.
* Seybold, Bryan, et al. "Dueling decoders: Regularizing variational autoencoder latent spaces." arXiv preprint arXiv:1905.07478 (2019).
* ...
Of course, also comparisons with these methods would be valuable.

**Q7 Justification For Your Score:**

I think the paper proposes a simple and effective idea for an important problem. The empirical evaluation could be extended to strengthen the paper (more baselines, studying the resulting learned representations).

**Q9 Complying With Reviewing Instructions:**

1: Yes.

---

### Decision · Program_Chairs · 2022-05-15

**Decision:**

Accept (Poster)

**Comment:**

Meta Review: The paper proposes a simple but effective method for preventing a posterior collapse in VAEs.

Quality: The work is technically sound. The experiments are adequate and support the claim.

Clarity: it is a well-written paper.

Originality:  The idea is not very new. Others have used MI to prevent posterior collapse, and this paper uses MI in a new way.

Significance: High, could be very useful, based on two reviewers.

One reviewer has asked a few questions about the novelty and the authors have provided reasonable answers.